# Posterior Oblique Square Decompression with a Three-Step Wanding Technique in Tubular Minimally Invasive Transforaminal Lumbar Interbody Fusion: Technical Report and Mid-Long-Term Clinical Outcomes

**DOI:** 10.3390/jcm11061651

**Published:** 2022-03-16

**Authors:** Takashi Tomita, Keita Kamei, Ryota Yamauchi, Takahiro Nakagawa, Hirotsugu Omi, Yoshiro Nitobe, Toru Asari, Gentaro Kumagai, Kanichiro Wada, Junji Ito, Yasuyuki Ishibashi

**Affiliations:** 1Department of Orthopaedic Surgery, Aomori Prefectural Central Hospital, Aomori 030-8553, Japan or k0bb_1_2ht_3_7ao_8@yahoo.co.jp (K.K.); omi0403@gmail.com (H.O.); jun2ito@yahoo.co.jp (J.I.); 2Department of Orthopaedic Surgery, Hirosaki University Graduate School of Medicine, Aomori 036-8562, Japan or ryoooota@hirosaki-u.ac.jp (R.Y.); n1t0bey0sh1r0@gmail.com (Y.N.); toru-asari@hirosaki-u.ac.jp (T.A.); gen722@hirosaki-u.ac.jp (G.K.); wadak39@hirosaki-u.ac.jp (K.W.); yasuyuki@hirosaki-u.ac.jp (Y.I.); 3Department of Orthopaedic Surgery, National Defense Medical College, Saitama 359-8513, Japan; tnakagawa0512@hotmail.com

**Keywords:** minimally invasive spinal treatment (MIST), minimally invasive spine stabilization (MISt), transforaminal lumbar interbody fusion (TLIF), tubular retractor

## Abstract

Although minimally invasive transforaminal lumbar interbody fusion (MIS-TLIF) is the most common procedure in minimally invasive spine stabilization (MISt), details of the technique remain unclear. This technical report shows the mid-long-term clinical outcomes in patients who underwent posterior oblique square decompression (POSDe) with the three-step wanding technique of tubular MIS-TLIF for degenerative lumbar disease. Tubular MIS-TLIF (POSDe) was performed on 50 patients (males, 19; age, 69.2 ± 9.6 years), and traditional open surgery was performed (OS) on 27 (males, 4; age, 67.9 ± 6.6 years). We evaluated the clinical outcomes using the Visual Analog Scale for back pain, Japanese Orthopedic Association (JOA) scores, and JOA Back Pain Evaluation Questionnaire. We also assessed the fusion rate using the Bridwell grading system with computed tomography or plain radiography for at least 2 years postoperatively. Although there was no significant difference in the improvement rate of JOA scores between the two groups, the mean operation time and blood loss were significantly lower with MIS-TLIF than with OS. In the tubular MIS-TLIF group, there were no cases of deep wound infection; four cases had a pseudarthrosis, two had dural injury, and three had cage retropulsion. We revealed good clinical outcomes in patients who underwent POSDe.

## 1. Introduction

It is widely accepted that Japan is developing as a super-aging society [1]; therefore, spine surgeons will have to manage many elderly patients in the near future. In such a condition, the concept of minimally invasive spine surgery (MISS) is very important and includes minimally invasive spine stabilization (MISt). The history of posterior lumbar interbody fusion (PLIF) was reported by Cloward in 1953 [2], and in 1982, Harms described the procedure as transforaminal lumbar interbody fusion (TLIF) [3]. During the spread of subsequent minimally invasive surgery (MIS), Foley in 2003 reported on minimally invasive transforaminal lumbar interbody fusion (MIS-TLIF) [4], which was applied for decompression in a tubular retractor and fixation with percutaneous pedicle screws (PPS). At present, with the spread of PPS, MIS-TLIF has become widely known as an indispensable technique in MISt for lumbar spinal disease [5,6,7,8,9]. Although there are many reports on MIS-TLIF [10,11,12,13,14,15], the authors adopted Foley’s technique for tubular surgery. In the case of lumbar spondylolisthesis, the unilateral approach for bilateral decompression is applied using a 26 mm diameter tubular retractor. There is also a report that standard fluoroscopies are used for intraoperative imaging in about 80% of MIS-TLIF cases [16]. This study aimed to provide a technical report and to report the mid-long-term clinical outcomes in patients who underwent the posterior oblique square decompression (POSDe) with the three-step wanding technique of tubular MIS-TLIF under standard fluoroscopic imaging for degenerative lumbar disease. Our hypothesis is that MIS-TILIF with POSDe is a safe and less invasive method compared with traditional open surgery, and is also economically beneficial.

## 2. Materials and Methods

### 2.1. Patient Selection

We retrospectively reviewed a total of 50 patients (males, 19; females, 31; age, 69.2 ± 9.6 years) who underwent one-level MIS-TLIF with POSDe; among these, five patients were operated on at L3/4, 40 at L4/5, and five at L5/S1. All surgeries were performed by a single surgeon (T Tomita) between 2017 and 2019, and all used the tubular retractor system. A control group (OS) was included, comprising a sequence of cases of TLIF with traditional open surgery performed by the same surgeon between 2007 and 2009. OS was performed on 27 patients (males, 4; females, 23; age, 67.9 ± 6.6 years); among these, three patients were operated on at L3/4, 21 at L4/5, and three at L5/S1 (Table 1).

All patients were followed up for at least 2 years. The mean follow-up period was 34 (range, 25–43) months. All patients underwent pre- and postoperative assessments via computed tomography (CT) and plain radiography. The indications for MIS-TLIF with POSDe were grade I and grade II lumbar spondylolisthesis. All patients had radiculopathy, including passing nerve root radiculopathy and/or exiting nerve root radiculopathy, and low back pain. Approximately half of the patients had a neurogenic claudication. In the preoperative imaging of CT or MRI, the facet gap was recognized in all patients, which verified the evidence of spinal instability [17,18]. This study was approved by the ethics committee of Aomori Prefectural Central Hospital (approval number: R3-AOBYODAI876GO), and informed consent was obtained from all patients before surgery.

### 2.2. Surgical Procedures

We started with the decompression from the more symptomatic side. The level of the intervertebral disc was marked, and a 30 mm skin incision was made at the medial boarder of the pedicles. After sequential dilation, the tubular retractor (26 mm) was inserted on the facet joint complex overlying the disc space (Figure 1a). The operation was performed under direct vision with illumination. The osteotomes (5 or 7 mm wide) were mainly used to extract the local bone for bone grafting during the decompression. Subsequently, most of the inferior articular process and pars interarticularis of the cranial vertebra were resected using an osteotome until the flavum tip appeared (Figure 1e–g). Next, we performed the first wanding of the tubular retractor to the opposite side on the lamina (Figure 1b). From here, it shifted to the opposite decompression (Figure 1h,i). To perform the opposite decompression more safely, it is advisable to resect the spinous process base on the entry side as much as possible such that a visual field can be obtained from above. This makes it easy to ensure opposite side decompression and avoids any potential complication concerning an opposite site nerve root. Bone decompression was thus performed by hollowing out the vertebral arch from the base of the spinous process to the opposite side, and it was possible to obtain a sufficient amount of the bone harvest if it was one intervertebral level at this stage. The final stage of decompression was on the caudal side. We performed the second wanding of the tubular retractor to the caudal side (Figure 1c). We used an osteotome at the base of the spinous process of the caudal lamina to break the cranial part of the spinous process less than 1 cm (Figure 1j), and resected the cranial part of the caudal lamina at this stage. The remaining flavum was also resected in approximately one lump, together with the attachment (Figure 1k). Here, it is important to detach the adhesion between the flavum and the dura while lifting the spinous process resected from the caudal side and proceed with the decompression. Finally, we performed the third wanding (Figure 1d); a so-called safety triangle zone was developed when an L-shaped partial excision was performed on the superior articular process of the caudal vertebra (Figure 1l,m). When manipulating a series of osteotomes, it is important to pay close attention to the direction and inclination of the blades to avoid dura and nerve root damage. We refer to this as “Posterior Oblique Square Decompression” (POSDe) because the decompression is finally performed with the wanding technique inside the retractor. We can accomplish the decompression of the neural elements directly through these processes. Next, the intervertebral disc was removed to the opposite side with a curette, and the harvested bone was crushed and inserted through a tubular device. Subsequently, a single cage was inserted. Following this operation, PPS were inserted using fluoroscopic guidance. The reduction and compression force were applied before tightening of the final construct (Figure 2).

### 2.3. Outcome Assessment

We investigated the surgical operative time and estimated blood loss for all patients. During both the perioperative and postoperative periods, the patients were followed by the surgeon for the presence of deep wound infections, pseudarthrosis, dural injury, cage migrations, and so on. For clinical outcomes, we used the Visual Analog Scale (VAS) for back pain and the Japanese Orthopedic Association (JOA) scores to assess the preoperative and postoperative (2 years post-surgery) pain and the disability status of the patients. Clinical outcomes of MIS-TLIF with POSDe were also evaluated using the JOA Back Pain Evaluation Questionnaire (JOABPEQ). According to radiological assessment, the fusion rates were assessed with the Bridwell grading system [6,19] via CT or plain radiography at the 2-year follow-up. All data were prospectively collected and retrospectively reviewed.

### 2.4. Statistical Analysis

Data are presented as means and standard deviations. We compared the MIS-TLIF and OS groups using the Mann–Whitney U test and Chi-squared test for qualitative and quantitative data. The indicators of dependent variables were POSDe and OS, while those of independent variables were VAS for back pain, JOA score, operative time, blood loss, and complications. A *p*-value < 0.05 was considered statistically significant. The statistical tool used for data input and statistical calculations was SPSS ver. 24.0J (SPSS Inc., Chicago, IL, USA).

## 3. Results

From the preoperative assessment to the 2-year follow-up, VAS for back pain improved from 64.3 to 12.1 with MIS-TLIF, and from 78.3 to 12.8 with OS (2-year follow-up, *p* = 0.05). The improvement rate was 59.4% with MIS-TLIF and 74.5% with OS, indicating a significant difference between the two groups. The JOA score improved from 12.1 to 26.0 with MIS-TLIF, and from 12.8 to 24.6 with OS (2-year follow-up, *p* = 0.05). The improvement rate was 82.2% with MIS-TLIF and 72.8% with OS; no significant difference was observed between the two groups. The mean operative time with MIS-TLIF (103.1 (range, 69–153) min) was significantly lower than that with OS (172.1 (range, 114–210) min). The mean blood loss with MIS-TLIF (74.2 (range, small amounts–280) mL) was significantly lower than that with OS (135.3 (range, 30–262) mL) (Table 2). In tubular MIS-TLIF with POSDe, the JOABPEQ significantly improved in each category at the 2-year follow-up (*p* = 0.05) (Figure 3). From the radiological assessment using the Bridwell fusion grading system, 33 cases had a grade 1 fusion, and 13 cases had a grade 2 fusion. There were two grade 3 cases and two grade 4 cases in the tubular MIS-TLIF group. In the OS group, 17 cases had a grade 1 fusion, and 6 cases had a grade 2 fusion; there were two grade 3 and two grade 4 cases, and no cases of deep wound infection in either group. Four cases (8.0%) had pseudarthrosis in the MIS-TLIF group, whereas two cases (7.4%) had pseudarthrosis in the OS group. There was no dural injury case in the OS group; however, two cases (4.0%) in the MIS-TLIF group had dural injury that required a dural recovery with fractionated plasma products or a dural suture, although they had good recovery. There were three cases (6.0%) of cage retropulsion in the MIS-TLIF group, and two (7.4%) in the OS group (Table 3).

## 4. Discussion

This study provided a technical report as well as the mid-long-term clinical outcomes of patients who underwent POSDe with the three-step wanding technique of tubular MIS-TLIF under standard fluoroscopic imaging for lumbar degenerative disease. We observed good clinical outcomes in patients who underwent the POSDe technique of tubular surgery. There are various surgical procedures for lumbar degenerative spondylolisthesis, including endoscopic TLIF [20,21,22], lateral lumbar interbody fusion (LIF) by indirect decompression [13,23,24,25,26,27], as well as MIS-TLIF. Currently, there are a few reports comparing MIS-TLIF with LIF [28,29,30,31,32]. MIS-TLIF can cope with various pathological conditions and is an indispensable procedure for cases that require direct decompression [28,30,33]. Therefore, this procedure should be advanced in the technique continuously. Minimally invasive techniques using a tubular retractor are one of the popular approaches to the spinal column [8,12,34,35,36,37]; however, the detailed technique remains unclear, especially how to decompress the neural elements in the tubular retractor effectively.

Our new procedure, posterior oblique square decompression (POSDe) with the three-step wanding technique, for tubular MIS-TLIF is considered to be a beneficial method. This procedure is based on the concept of achieving bilateral spinal decompression through a unilateral approach, which was first applied to open lumbar surgery in the 1990s [38,39]. The unilateral laminotomy for bilateral decompression was applied to the minimal invasive spinal procedure for the treatment of lumbar spinal canal stenosis in 2002 [34,40,41]. Foley has applied this concept to tubular surgery [4], additionally mentioning that the benefit of MIS is achieving the goals of minimally invasive fusion procedures by reducing the approach-related morbidity associated with the traditional fusion procedure. We adopted Foley’s technique for tubular surgery in this study. Holly emphasized the merit of using the tubular retractor, which minimizes approach-related morbidity [35]. Inanami mentioned microendoscope-assisted PLIF [42], wherein the tubular retractor to various directions can be changed (this is difficult under a microscope). In our POSDe technique, it is possible to change the tubular retractor with three-step wanding, and decompression of the opposite side becomes easy. Essentially, the point of using sequential dilators in tubular surgery is to minimize irreversible changes, such as crushing of the paraspinal muscles and ischemia [43]. The 26 mm diameter of the tubular retractor is an appropriate size to complete the one-level decompression and insert the cage safely. The operative time and blood loss with MIS-TLIF are lower than that those reported previously with OS [3,4,5,8,36,42,44]. Regarding the operative time of MIS-TLIF, the time for decompression becomes a rate-limiting step. The merit of using an osteotome in POSDe enables the operative time to be reduced. The patients revealed more improvements in their postoperative VAS, JOA scores, and JOABPEQ compared with their preoperative scores.

Park reported that solid fusion could be accomplished via minimally invasive techniques in the same way as the traditional open procedure [5]. Although harvested bone was used, our study also revealed a satisfactory fusion rate. The rate of dural injury was 4.0% (2/50) in our study, whereas Park reported 0% among 32 cases [5]; thus, our rate of dural injury was high, suggesting a learning curve for this procedure [45]. As Nandyala et al. mentioned regarding learning curves, it is important to obtain additional training and more experience to ensure satisfactory outcomes of the technique [37].

The infection rate was 0% in our study; this is due to the shorter operative time and small skin incision [46]. One of the features of the POSDe technique is the use of an osteotome, allowing enough autologous bone to be harvested to one-level intervertebral disc space; we therefore did not need to use an artificial bone, which leads to positive economic effects and a decreased medical bill. Not using artificial bone resulted in a cost reduction of about JPY 80,000 per one-level MIS-TLIF [47]. Harrop et al. mentioned the importance of the cost-effective treatment [48]. It is necessary to pay attention to the economic benefits [49,50,51], and this is an important aspect of MIST.

Lener et al., reported that standard fluoroscopy is used for intraoperative imaging in approximately 80% of MIS-TLIF cases [16]. POSDe under standard fluoroscopy appears to be very effective and widely useable at this point. However, the development of a navigation system and virtual reality is expected to support the concept of POSDe with the three-step wanding technique of tubular MIS-TLIF in the near future.

The study had several limitations. First, POSDe using an osteotome may have a learning curve. Second, the number of conventional open TLIF procedures was smaller than that of MIS-TLIF, as we had completely shifted to MIS-TLIF; therefore, we could not use the JOABPEQ for OS and match the sex ratio between the two groups.

## 5. Conclusions

We revealed good mid-long-term clinical outcomes in patients who underwent the POSDe technique of tubular surgery, which was found to be a safe and minimally invasive technique for the treatment of degenerative lumbar disease cases compared with traditional open surgery. The advantages of using the POSDe technique were as follows: (I) higher operability (three-step wanding to various angles), (II) shorter operative time, (III) lower blood loss, and (IV) adequate local bone harvesting. These advantages are considered to decrease soft tissue damage, operative time, and medical bills. Although we still need to follow the patients in the long term, this appears to be safe and suitable for tubular MIS-TLIF in patients with degenerative lumbar disease.

## Figures and Tables

**Figure 1 jcm-11-01651-f001:**
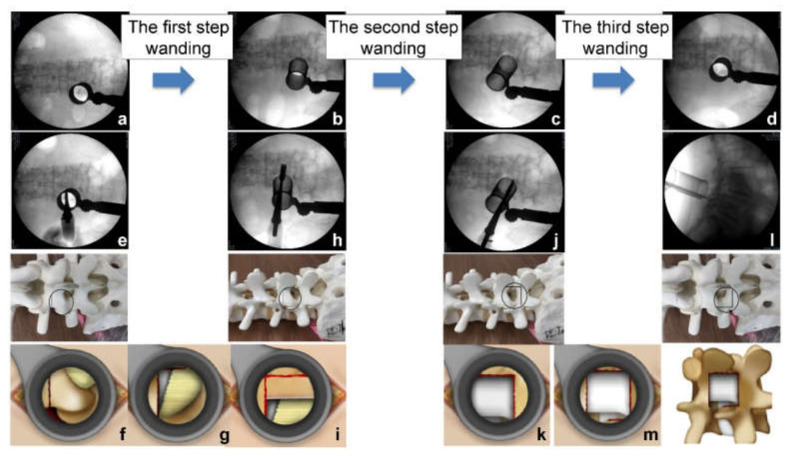
Posterior oblique square decompression with three-step wanding technique. (**a**) Tubular retractor insertion. (**b**) The first wanding to the opposite side. (**c**) The second wanding to the caudal side. (**d**) The third wanding. (**e**,**f**) The inferior articular process and pars interarticularis resection. (**g**) The identification of the flavum tip and the dura. (**h**,**i**) The opposite side decompression. (**j**) The cranial part of inferior vertebra resection. (**k**) The remaining flavum resection. (**l**) The L-shaped excision on the superior articular process of the caudal vertebra. (**m**) Completion of posterior oblique square decompression.

**Figure 2 jcm-11-01651-f002:**
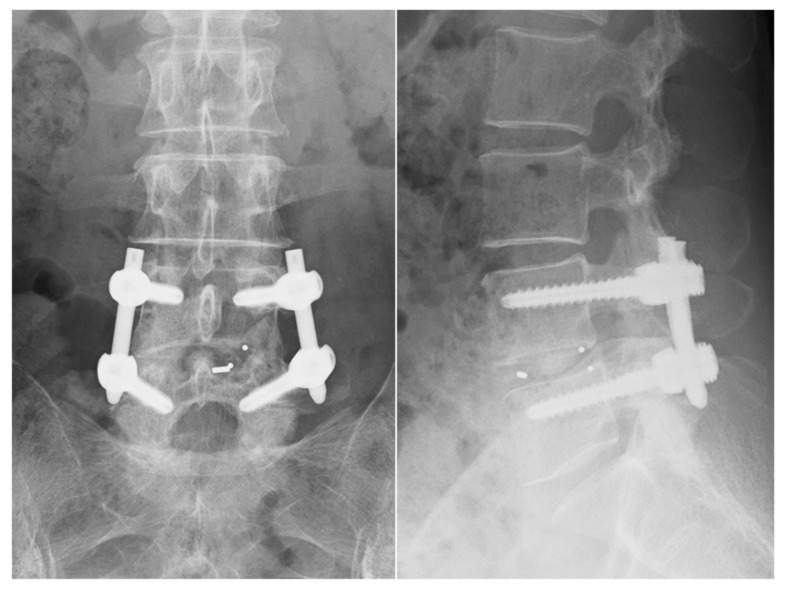
Postoperative X-ray after MIS-TLIF (coronal and lateral views). MIS-TLIF, minimally invasive transforaminal lumbar interbody fusion.

**Figure 3 jcm-11-01651-f003:**
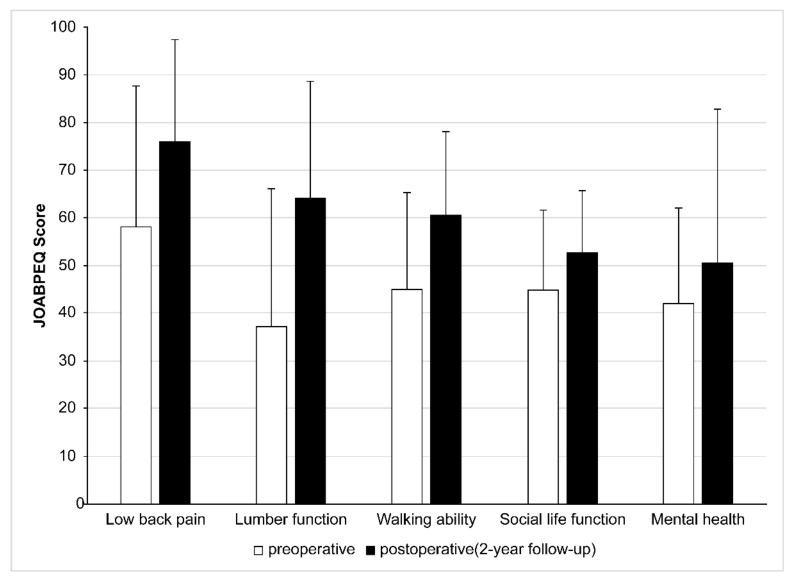
JOABPEQ in POSDe with the three-step wanding technique. JOABPEQ in POSDe improved in each category significantly (2-year follow-up, *p* = 0.05). JOABPEQ, Japanese Orthopedic Association Back Pain Evaluation Questionnaire; POSDe, posterior oblique square decompression.

**Table 1 jcm-11-01651-t001:** Patient characteristics.

Characteristics	POSDe	OS	*p*-Value
No. of cases	50	27	
Age (year), means ± SD (range)	69.2 ± 9.6 (46–89)	67.9 ± 6.6 (49–78)	0.361
Sex (male:female)	19:31	4:23	0.0397 ^#^
No. of levels treated			1.000
L3/4	5	3	
L4/5	40	11	
L5/S1	5	3	

^#^ Chi-squared test. POSDe, posterior oblique square decompression; OS, control group

**Table 2 jcm-11-01651-t002:** Clinical outcomes.

Characteristics	POSDe	OS	*p*-Value
VAS for back pain			
Preop.	64.3	78.3	<0.0001 *
Postop. (2 years)	12.1	12.8	<0.0001 *
JOA score			
Preop.	12.1	12.8	0.423
Postop. (2 years)	26.0	24.6	0.0514
Operation time (min)	103.1	172.1	<0.0001 *
Blood loss (mL)	74.2	135.3	<0.0001 *

POSDe, posterior oblique square decompression; OS, control group; VAS, visual analog scale; JOA, Japanese Orthopedic Association; * Mann–Whitney U test.

**Table 3 jcm-11-01651-t003:** Complications.

Complications	POSDe N (%)	OS N (%)	*p*-Value
Deep wound infection	0 (0)	0 (0)	1.000
Pseudoarthrosis	4 (8)	2 (7.4)	1.000
Dural injury	2 (4)	0 (0)	0.55
Cage retropulsion	3 (6)	2 (7.4)	1.000

POSDe, posterior oblique square decompression; OS, control group.

## Data Availability

Not applicable.

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
