# Peer review of "Posterior Oblique Square Decompression with a Three-Step Wanding Technique in Tubular Minimally Invasive Transforaminal Lumbar Interbody Fusion: Technical Report and Mid-Long-Term Clinical Outcomes"

_jcm, 2022, doi:10.3390/jcm11061651_

Round 1
Reviewer 1 Report
Authors present their experience with the Posterior Oblique Square Decompression technique for treatment of symptomatic spondylolisthesis. Authors report excellent results and compare them to traditional OS??.
The article is certainly interesting. There are a few issues that could be improved
- What exactly is OS? This is never defined other than saying it is the "control group". What does this stand for? Open surgery? this needs to be clearly defined in the abstract and throughout the paper
- There are a lot of abbreviations in the paper, is it possible to remove some of these? These are also very confusing (for example, MIST vs MISt?, or OS vs POSDe?) I would try to minimize the use of these as much as possible
- How would one choose whether to do a POSDe or OS? How did the surgeon decide which technique? Was this just random? Or based on certain imaging or clinical features?
- How many patients had neurogenic claudication? how many had only radiculopathy? any other symptoms?
- What are some potential complications that could be encountered and how can one overcome these? Inadequate decompression? CSF leak? Any others? It would be useful for surgeon learning this technique for the first time
Author Response
Reviewer 1
We deeply appreciate the reviewers’ thoughtful and constructive comments, which provided important scientific suggestions that helped us improve our manuscript.
- What exactly is OS? This is never defined other than saying it is the "control group". What does this stand for? Open surgery? this needs to be clearly defined in the abstract and throughout the paper
Our response
We appreciate your suggestion. We corrected the abstract to define OS as follows and to use OS throughout the paper.
P1
This technical report shows the mid-long-term clinical outcomes in patients who underwent the posterior oblique square decompression (POSDe) with three-step wanding technique of tubular MIS-TLIF for degenerative lumbar disease. Tubular MIS-TLIF (POSDe) was performed on 50 patients (males, 19; age, 69.2±9.6 years), and traditional open surgery (OS) on 27 (males, 4; age, 67.9±6.6 years).
- There are a lot of abbreviations in the paper, is it possible to remove some of these? These are also very confusing (for example, MIST vs MISt?, or OS vs POSDe?) I would try to minimize the use of these as much as possible 

Our response
We deeply appreciate your comments. We used MISS in the Introduction and Discussion sections and used MISt in the Abstract and Introduction. Both words are the key words used in the manuscript. MIST includes MISt, which is the surgical technique.
There are a lot of abbreviations because this a comparative study between POSDe and OS, although I tried to minimize the use of these as much as possible.
- How would one choose whether to do a POSDe or OS? How did the surgeon decide which technique? Was this just random? Or based on certain imaging or clinical features? 

Our response
We deeply appreciate your suggestion. The indication of TLIF for spondylolisthesis is the same in both POSDe and OS. This is a retrospective cohort study comparing two consecutive groups of patients who underwent TLIF for one-level spondylolisthesis. Cohort A (2007-2009) included 27 patients receiving traditional open surgery (OS). Cohort B (2017-2019) included 50 patients receiving MIS-TLIF with POSDe. Therefore, we added the period of each surgery in the Material and methods. Moreover, one of our indications of TLIF was the facet gap on the preoperative imaging. Therefore, we added this point in the Material and methods as follows.
P2
We retrospectively reviewed a total of 50 patients (males, 19; females, 31; age, 69.2±9.6 years) who underwent one-level MIS-TLIF with POSDe; among these, five patients were operated on at L3/4, 40 at L4/5, and five at L5/S1. All surgeries were performed by a single surgeon (T Tomita) between 2017 and 2019, and all used the tubular retractor system. A control group (OS) was included, comprising a sequence of cases of TLIF with traditional open surgery performed by the same surgeon between 2007 and 2009. OS was performed on 27 patients (males, 4; females, 23; age, 67.9±6.6 years); among these, three patients were operated on at L3/4, 21 at L4/5, and three at L5/S1 (Table 1).
In the preoperative imaging of CT or MRI, the facet gap was recognized in all patients, which verified the evidence of spinal instability [42,43].
- How many patients had neurogenic claudication? how many had only radiculopathy? any other symptoms? 

Our response
We deeply appreciate your suggestion. All patients had a radiculopathy, including passing nerve root radiculopathy and exiting nerve root radiculopathy, and low back pain. Approximately half of patients had neurogenic claudication. Thus, we added the description of the symptoms in the Material and methods as follows.
P2
The indications for MIS-TLIF with POSDe were grade I and grade II lumbar spondylolisthesis. All patients had radiculopathy, including passing nerve root radiculopathy and/or exiting nerve root radiculopathy, and low back pain. Approximately half of patients had a neurogenic claudication.
- What are some potential complications that could be encountered and how can one overcome these? Inadequate decompression? CSF leak? Any others? It would be useful for surgeon learning this technique for the first time 

Our response
We thank you for your very important comments. It is very important to pay attention to an opposite site passing nerve root. To avoid injury during decompression, the view of the operative field should be sufficient and we should protect the opposite site passing nerve root. We added the avoidance of potential complications concerning an opposite site nerve root to the revised manuscript. To avoid an inadequate decompression, we also mentioned the point of decompression. Regarding a CSF leak, we performed a dural recovery with fractionated plasma products or a dural suture.
P3
To perform the opposite decompression more safely, it is advisable to resect the spinous process base on the entry side as much as possible such that a visual field can be obtained from above. This makes it easy to ensure opposite side decompression and avoids any potential complication concerning about an opposite site nerve root.
Reviewer 2 Report
Introduction: Please provide a research hypothesis. Please specify the dependent and independent variables and the indicators of these variables.
Materials and methods: Please describe in detail the method of selecting people for research.
Discussion:
The discussion needs to be expanded. Cite more similar studies.
Conclusions: Please indicate the practical importance of this research.
Author Response
Response to Reviewer 2
We deeply appreciate the reviewers’ thoughtful and constructive comments, which provided important scientific suggestions that helped us improve our manuscript.
Introduction: Please provide a research hypothesis.
Our response
We deeply appreciate your suggestion. We added our hypothesis in the introduction as follows.
P2
This study aimed to provide a technical report and to report the mid-long-term clinical outcomes in patients who underwent the posterior oblique square decompression (POSDe) with three-step wanding technique of tubular MIS-TLIF under standard fluoroscopic imaging for degenerative lumbar disease. Our hypothesis is that MIS-TILIF with POSDe is a safe and less invasive method compared with traditional open surgery, and is also economically beneficial.
Please specify the dependent and independent variables and the indicators of these variables.
Our response
We deeply appreciate your suggestion. We specified the dependent and independent variables as follows.
P4
2.4. Statistical analysis
Data are presented as means and standard deviations. We compared the MIS-TLIF and OS groups using the Mann-Whitney U test and Chi-squared test for qualitative and quantitative data. The indicators of dependent variables were POSDe and OS, while those of the independent variables were VAS for back pain, JOA score, operative time, blood loss, and complications. A p-value less <0.05 was considered statistically significant. The statistical tool used for data input and statistical calculations was SPSS ver.24.0J (SPSS Inc., Chicago, IL, USA).
Materials and methods: Please describe in detail the method of selecting people for research.
Our response
We deeply appreciate your suggestion. This study is a retrospective cohort study comparing two consecutive groups of patients who underwent TLIF for one-level spondylolisthesis. Cohort A (2007-2009) included 27 patients receiving traditional open surgery (OS) and Cohort B (2017-2019) included 50 patients receiving MIS-TLIF with POSDe. Therefore, we described the methods of selecting people for research and added the period of each surgery in the Material and methods as follows.
P2
We retrospectively reviewed a total of 50 patients (males, 19; females, 31; age, 69.2±9.6 years) who underwent one-level MIS-TLIF with POSDe; among these, five patients were operated on at L3/4, 40 at L4/5, and five at L5/S1. All surgeries were performed by a single surgeon (T Tomita) between 2017 and 2019, and all used the tubular retractor system. A control group (OS) was included, comprising a sequence of cases of TLIF with traditional open surgery performed by the same surgeon between 2007 and 2009. OS was performed on 27 patients (males, 4; females, 23; age, 67.9±6.6 years); among these, three patients were operated on at L3/4, 21 at L4/5, and three at L5/S1 (Table 1).
Discussion: The discussion needs to be expanded. Cite more similar studies.
Our response
We deeply appreciate your suggestion. We cited and added 20 references, including similar studies, and edited the discussion. We also expanded the discussion concerning the technical merit and economical effectiveness.
Conclusions: Please indicate the practical importance of this research.
Our response
We deeply appreciate your suggestion. We emphasized the advantages of POSDe compared with a traditional open surgery as follows.
P7
We revealed the good clinical outcomes in patients who underwent the POSDe technique of tubular surgery, which was found to be a safe and minimally invasive technique for the treatment of degenerative lumbar disease cases compared with a traditional open surgery.

Reviewer 3 Report
MIS TLIF already too many research paper and procedure is explained in many papers.
If there is any relavance and long time follow up can be useful,
tecinical report is not valuable anymore and nothing new finding compare to other study
Author Response
Response to Reviewer 3
We deeply appreciate the reviewers’ thoughtful and constructive comments, which provided important suggestions that helped us improve our manuscript.
MIS TLIF already too many research paper and procedure is explained in many papers.
If there is any relavance and long time follow up can be useful,
tecinical report is not valuable anymore and nothing new finding compare to other study
Our response
We appreciate your important comments.
I apologize; however, I would like to introduce MISt in MIST through MIS-TLIF. This manuscript is not only a technical report but also presents the history and concept of MIS-TLIF and economical merits by POSDe. It is believed that the concern about medical bill and cost effectiveness is increasing. I believe that this may be one of the new findings.
On the other hand, XLIF and OLIF are widely available worldwide. According to direct decompression, MIS-TLIF is one of the indispensable procedures. This special technique should be preserved and advanced continuously.
POSDe is one of the refine techniques, especially on how to decompress the neural elements in the tubular retractor effectively. Its operative time is very short, and the blood loss amount is less than those described in previous reports. Regarding the operative time of MIS-TLIF, the time for decompression becomes a rate-limiting step. The merit of using an osteotome in POSDe is that it results in a very short operative time.
Finally, the mean follow-up period was 34 (range, 25–43) months in our study, and we changed the clinical outcomes to mid-long-term clinical outcomes in the title and throughout the paper.

Round 2
Reviewer 3 Report
I didn`t find any special valuable academic point to store this paper Metrix retactor using posterior oblique square decompression for disc approach is not new technique. Also MISS TLIF versus open PLIF already published a lot